# AReaL: A Large-Scale Asynchronous Reinforcement Learning System for Language Reasoning

**Wei Fu**[12]*, **Jiaxuan Gao**[1], **Xujie Shen**[2], **Chen Zhu**[2], **Zhiyu Mei**[12],
**Chuyi He**[2], **Shusheng Xu**[12], **Guo Wei**[2], **Jun Mei**[2], **Jiashu Wang**[3],
**Tongkai Yang**[2], **Binhang Yuan**[3], **Yi Wu**[1]

[1] IIIS, Tsinghua University, [2] Ant Group, [3] HKUST
fuwth17@gmail.com, jxwuyi@gmail.com

## Abstract

Reinforcement learning (RL) has become a trending paradigm for training large language models (LLMs), particularly for reasoning tasks. Effective RL for LLMs requires massive parallelization and poses an urgent need for efficient training systems. Most existing large-scale RL systems for LLMs are synchronous by alternating generation and training in a batch setting, where the rollouts in each training batch are generated by the same (or latest) model. This stabilizes RL training but suffers from severe system-level inefficiency. Generation must wait until the longest output in the batch is completed before model update, resulting in GPU underutilization. We present AReaL, a *fully asynchronous* RL system that completely decouples generation from training. Rollout workers in AReaL continuously generate new outputs without waiting, while training workers update the model whenever a batch of data is collected. AReaL also incorporates a collection of system-level optimizations, leading to substantially higher GPU utilization. To stabilize RL training, AReaL balances the workload of rollout and training workers to control data staleness, and adopts a staleness-enhanced PPO variant to better handle outdated training samples. Extensive experiments on math and code reasoning benchmarks show that AReaL achieves **up to 2.77× training speedup** compared to synchronous systems with the same number of GPUs and matched or even improved final performance. The code of AReaL is available at https://github.com/inclusionAI/AReaL/.

## 1 Introduction

Reinforcement learning (RL) has emerged as a new scaling paradigm for enhancing the capabilities of large language models (LLMs) by enabling thinking abilities [52]. Given a prompt, RL allows an LLM to generate thinking tokens before outputting a final answer, enabling *test-time scaling* [29, 47]. These thinking LLMs are named Large Reasoning Models (LRMs) and have been shown to have particularly strong capabilities on challenging reasoning problems, such as math [9, 5, 20], coding [3, 14, 15], logic puzzles [22, 34], and agentic tasks [23, 57].

Effective RL training often requires massive parallelization to derive a large batch of rollouts for sufficient exploration, which is the key to obtaining optimal model performance. For example, popular RL algorithms, such as PPO [42] and GRPO [43], often require an effective training batch of thousands of outputs [60, 61, 53]. Moreover, an LRM can generate tens of thousands of thinking tokens for each input prompt [6], further posing an urgent need for an efficient training system to run RL training on a large scale.

---

*This work was supported by Ant Group Research Intern Program

However, developing an efficient large-scale RL system is challenging. An RL system needs to frequently switch between LLM generation and training, which can introduce significant system overhead without careful optimizations. For LRMs, the output length of the training model varies significantly for different prompts throughout the RL process, which results in an ever-changing workload for both generation and training. This characteristic often triggers idle time in high-performance hardware, leading to a waste of computation. Furthermore, classical large-scale RL algorithms like PPO or GRPO typically require on-policy training data, i.e., samples generated by the latest model, to ensure the best model performance, which poses additional system challenges.

Consequently, most existing large-scale RL systems are designed in a fully synchronous manner [27, 11, 45, 44] by strictly alternating between LLM generation and training, which ensures that the LLM is always trained on the latest outputs for the best practical performance. In such a synchronous design, the generation step must wait until the finish of the longest output within a batch. Due to the varying output lengths for LRMs, a synchronous RL system suffers from severe training inefficiency. Very recently, there have also been attempts to explore parallel generation and training [30, 24, 49]. These works use outputs generated from a previous model version to update the current model. For the best performance, the model version used for rollout generation is limited to only one or two steps older. However, all these systems still follow a batched generation setting, where all the samples within a training batch are from the same model version. Accordingly, the issue of system inefficiency during the generation phase still remains unaddressed.

To fundamentally resolve the issues in system design, we develop AREAL, a fully Asynchronous RL training system for LRMs that completely decouples generation from training without hurting the final performance. AREAL runs LLM generation in a streaming manner, where each rollout worker continuously generates new outputs without waiting, leading to high GPU utilization. Meanwhile, the trainer workers in AREAL run parallel model updates whenever a training batch is obtained from the rollout workers. Once the model is updated, we synchronize the model weights in each rollout worker. In such an asynchronous design, each training batch of AREAL may contain samples generated by different model versions. Therefore, AREAL incorporates a modified objective of the PPO algorithm, which can leverage samples generated from much older model versions without any performance drop. AREAL also conducts a data filtering process to ensure the staleness of each training sample is well controlled. In addition, AREAL also introduces several system-level optimizations, including interruptible rollout workers, dynamic batching for variable-length outputs, and parallel reward service, which further improve the overall training throughput.

We evaluate AREAL on challenging mathematical reasoning and code generation tasks using models up to 32B parameters. Compared to state-of-the-art synchronous systems, AREAL achieves up to $2.57\times$ higher training throughput and linear scaling efficiency up to 512 GPUs. Crucially, this acceleration *even comes with improved solution accuracy on these tasks*, illustrating that AREAL delivers significant efficiency gains without sacrificing (and indeed enhancing) model performance.

sectionRelated Work

**RL for LLMs** Reinforcement learning (RL) has emerged as the predominant paradigm for enhancing the reasoning capabilities of Large Language Models (LLMs) [31, 32]. Existing RL approaches typically focus on tasks with well-defined reward functions, including mathematical reasoning [9], coding [14, 15], scientific problem solving [39, 36], and tool use [57]. During training, models learn to reason by progressively extending the length of chain-of-thought trajectories [52, 6]. Recent open-source initiatives have demonstrated significant success in improving model capabilities through smaller distilled models [24, 25]. Our work builds upon this research direction, distinguishing itself from preference-based RLHF [33] and zero-shot reasoning approaches [60, 61, 12] that attempt to acquire reasoning skills from pre-trained models without task-specific fine-tuning.

**Asynchronous RL** The decoupled asynchronous RL architecture [21, 8, 26], combined with corresponding algorithmic innovations [7, 16], has achieved remarkable success in game applications [2, 51]. Although similar asynchronous approaches have been explored for LLM training, they typically focus on short-context settings [30, 1, 40] (e.g., RLHF) or one/two-step generation-training overlap [24, 48]. Our work extends these studies and provides a more flexible trade-off between staleness and training speed, as we will show in Section 4. In contrast to concurrent work [64] that maximizes system-level efficiency, we adopt an algorithm-system co-design approach that provides both an expressive system and a practical algorithm implementation. Our interruptible generation technique is conceptually similar to partial rollout [17] in synchronous RL systems. Instead of

setting a fixed length budget, AREAL dynamically interrupts generation while maintaining consistent training batch sizes through buffering, thus preserving the stability of PPO. Compared with prior methods [40, 30], our algorithmic innovation in the asynchronous setting can endure higher data staleness and remains compatible with interruptible generation.

**LLM Training and Inference** Our work focuses on dense transformer models [50]. The RL training primarily consists of generation (inference) and training phases. Generation involves auto-regressive decoding, which requires efficient KV cache management [63, 18] and optimized decoding kernels [58]. Training requires careful orchestration of data, tensor, and pipeline parallelism strategies [38, 46, 62]. While conventional synchronous systems execute generation and training sequentially on the same hardware resources, they require different optimal parallelization strategies. Recent work has proposed context switching [19, 17] or weight resharding [45, 27] techniques to address this mismatch. AREAL advances beyond synchronous RL systems by decoupling generation and training, completely eliminating resharding overhead from the critical training path.

## 2 Background

### 2.1 Preliminaries about RL Training

**RL Formulation and PPO** We formulate our problem within the Markov Decision Process (MDP) framework [37], defined by the tuple $\langle \mathcal{S}, \mathcal{A}, r, P, \gamma, H \rangle$. Here, $\mathcal{S}$ represents the state space, $\mathcal{A}$ the action space, $P$ the transition model, $r : \mathcal{S} \times \mathcal{A} \to \mathbb{R}$ the reward function, $\gamma$ the discount factor, and $H$ the horizon. The LRM implements a parameterized policy $\pi_\theta : \mathcal{S} \to \mathcal{A}$ where each action $a_t \in \mathcal{A}$ corresponds to a text token from the vocabulary. The state $s_t \in \mathcal{S}$ consists of a question $s_1 = q$ followed by previously generated response tokens $(a_1, .., a_{t-1})$, with deterministic transitions $s_{t+1} = \text{concat}(s_t, a_t)$. Given a question distribution $\mathcal{D}$, we optimize the objective:

$$J(\theta) = \mathbb{E}_{q \sim \mathcal{D}, a_t \sim \pi_\theta(\cdot|q, a_{<t})} \left[ \sum_{t=1}^{H} \gamma^{t-1} r(s_t, a_t) \right]. \tag{1}$$

Following common practice [6, 25], we use a rule-based reward function that only provides non-zero feedback on the final action, indicating answer correctness, and set $\gamma = 1$. We optimize this objective using Proximal Policy Optimization (PPO) [42]:

$$J_{\text{PPO}}(\theta) = \mathbb{E}_{q \sim \mathcal{D}, a_t \sim \pi_{\text{old}}(\cdot|q, a_{<t})} \left[ \sum_{t=1}^{H} \min \left( u_t(\theta) \hat{A}(s_t, a_t), \text{clip}\left(u_t(\theta), 1 - \epsilon, 1 + \epsilon\right) \hat{A}(s_t, a_t) \right) \right], \tag{2}$$

where $u_t(\theta) = \frac{\pi_\theta(a_t|s_t)}{\pi_{\text{old}}(a_t|s_t)}$ denotes the importance ratio and $\hat{A}(s_t, a_t)$ represents the estimated advantage [41]. Following standard practices in RL [42, 33], we divide the global batch into minibatches for sequential parameter updates.[2]

**Distributed Systems for LRM Training** Our work focuses on enhancing reasoning capabilities for LRMs after Supervised Fine-Tuning (SFT), distinct from approaches that incentivize reasoning in pre-trained base models [6]. LRMs after SFT produce long reasoning sequences (e.g., 32K tokens) and usually require large global batch sizes (e.g., 128 prompts with 16 responses each) for stable RL training [6, 25, 24, 60, 61]. In *synchronous RL systems*, two phases are iteratively executed: generation (rollout) and training. The generation phase uses the latest model parameters to produce multiple reasoning traces for each query in the training batch. The training phase then updates the model parameters based on the generated trajectories. These phases execute iteratively on the same GPUs.

### 2.2 Motivation for Asynchronous RL System

We identify two essential limitations in synchronous RL systems:

**Inference devices are underutilized**. As shown in Figure 1 (left), generation must wait for the longest sequence to complete before training can begin. This leads to non-uniform decoding length across GPUs, which underutilizes GPU compute resources.

---

[2]This differs from gradient accumulation, which performs a single update across minibatches.

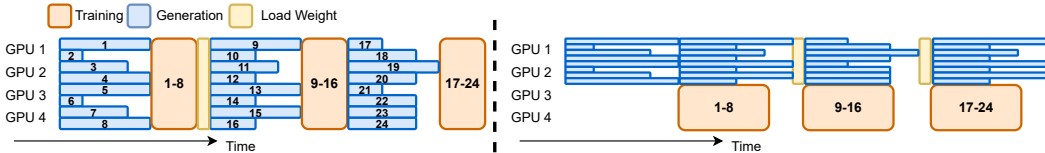

Figure 1: Execution timeline of a synchronous (left) and a one-step overlap (right) RL system showing underutilized inference devices.

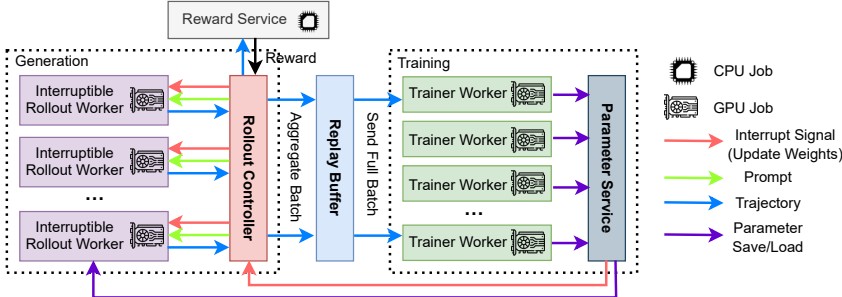

Figure 2: The AREAL architecture featuring asynchronous generation and training components.

**Scalability is poor in synchronous RL systems**. Synchronous systems distribute generation across all devices, reducing the per-GPU decoding batch size. This pushes the decoding process into a memory-IO-bound regime [4, 28] where additional devices fail to improve throughput.

## 3  System Architecture

The limitations identified in Section 2.2 motivate our design of a system that fully decouples generation and training across separate GPU clusters. This system should be hardware-efficient, scalable, and equipped with the flexibility for a customized RL workflow. We implement these principles in AREAL, an asynchronous RL system specifically designed for efficient large-scale LRM training.

### 3.1  System Overview

Figure 2 presents the architecture and data flow of AREAL. The system comprises 4 core components:

**Interruptible Rollout Worker** handles two types of requests: (1) The `generate` request generates responses given prompts. (2) The `update_weights` request interrupts all ongoing generations and loads parameters of new versions. Upon the interruption, the rollout workers discard KV caches computed by old weights, and re-compute them using the new weights. Afterwards, the rollout workers continue to decode the unfinished sequences until the next interruption or termination. We emphasize that such interruptions and in-flight weight updates would result in trajectories composed of segments produced by different model versions. This introduces a novel algorithmic challenge, which will be addressed in Section 4.

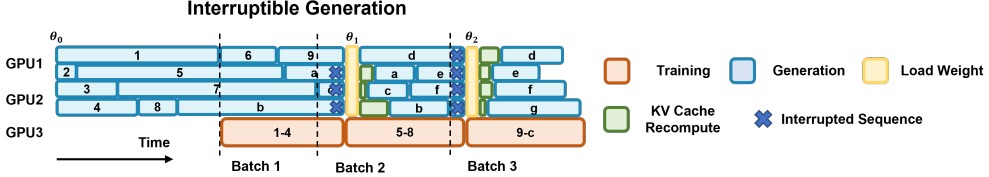

Figure 3: Illustration of generation management in AREAL. Vertical lines show the ready time for the next step training. Blue crosses show the interrupted requests when new parameters arrive.

**Reward Service** evaluates the accuracy of the responses generated by the model. For example, in the coding task, this service extracts the code and executes unit tests to verify its accuracy.

**Trainer Workers** continuously sample from the replay buffer, accumulating data until reaching the configured training batch size. They then perform PPO updates and store the resulting parameters in distributed storage. To ensure data freshness, data from the replay buffer is used only once.

**Rollout Controller** serves as a critical bridge between the rollout workers, reward service, and the model workers. During the training process, it reads data from the dataset and invokes the rollout worker's `generate` request. The received response is then sent to the reward service to obtain the reward. The trajectory, along with the reward, is stored in the replay buffer, waiting to be trained by the model worker. After the model worker updates the parameters, the controller calls the rollout worker's `update_weight`. We illustrate the generation and training management in Figure 3. This asynchronous pipeline ensures continuous full utilization of both generation and training resources.

### 3.2 Algorithmic Challenges

While the asynchronous system design offers significant acceleration through improved device utilization, it introduces several technical challenges that require algorithmic considerations.

**Data Staleness** Due to the asynchronous nature of AREAL, each training batch contains data from multiple prior policy versions. Prior works on asynchronous RL training systems have demonstrated that such staleness can degrade learning performance in both RLHF [30] and game environments [2]. Data staleness leads to a distribution gap between the training data and the latest model. In asynchronous RL training for LRMs, this issue could be even more severe for long trajectories due to extended decoding time.

**Inconsistent Policy Versions** As discussed in Section 3.1, the generated trajectories may involve segments produced by different policy versions. This inconsistency fundamentally violates the formulation of standard PPO in Eq. 2 that assumes all actions are generated by a single policy $\pi_{\text{old}}$.

In the following section, we detail our technical innovations for overcoming these challenges while preserving the efficiency advantages of an asynchronous system.

## 4 Addressing the Algorithmic Challenges in AREAL

### 4.1 Staleness-Aware Training

To avoid the performance drop due to training on data with extremely high staleness, we introduce a hyperparameter $\eta$ representing *the maximum permitted staleness in each training batch* for staleness-aware training. In particular, when $\eta = 0$, our system degenerates to synchronous RL with all training samples generated by the current policy. We implement the staleness control in our system by dynamically controlling the throughput of the generation requests sent to the generation servers. Given the current policy version $i$, the total number of generated trajectories $N_r$, and the training batch size $B$ for each training step, we enforce the following formula whenever submitting new generation requests:

$$\lfloor (N_r - 1)/B \rfloor \leq i + \eta. \tag{3}$$

We also prioritize older trajectories from the data buffer to form a training batch. In our system implementation, the rollout controller tracks both the generated samples $N_r$ and policy version $i$ from the parameter server. It rejects new generation requests that may violate the staleness constraint.

Note that this rate-limiting protocol is a simple yet effective design choice in practice. However, when $\eta$ is too small, the generation throughput can be slowed down when some extremely long trajectories are being generated. Therefore, we empirically suggest adopting a large staleness-control parameter $\eta$ for the best system throughput. This system-wide practice also motivates us to apply an enhanced algorithm that can make effective use of more stale data for RL training.

### 4.2 Decoupled PPO Objective

We apply a decoupled PPO objective [10] that disentangles the *behavior policy* and the *proximal policy*. The behavior policy $\pi_{\text{behav}}$ represents the policy used for sampling trajectories and the

proximal policy $\pi_{\mathrm{prox}}$ is a proximal policy serving as a recent target to regularize the update of $\pi_\theta$. By applying importance sampling on the sampled trajectories, we derive a decoupled PPO objective suitable for asynchronous RL training:

$$
J(\theta) = \mathbb{E}_{q \sim \mathcal{D}, a_t \sim \pi_{\mathrm{behav}}} \left[ \sum_{t=1}^{H} \min\left( \underbrace{\boxed{\frac{\pi_\theta}{\pi_{\mathrm{behav}}}}}_{\text{Importance Ratio}} \hat{A}_t, \overbrace{\frac{\pi_{\mathrm{prox}}}{\pi_{\mathrm{behav}}} \mathrm{clip}\left( \underbrace{\boxed{\frac{\pi_\theta}{\pi_{\mathrm{prox}}}}}_{\text{Trust Region Center}}, 1-\epsilon, 1+\epsilon \right) \hat{A}_t}^{\text{Importance Ratio}} \right) \right]
\tag{4}
$$

$$
= \mathbb{E}_{q \sim \mathcal{D}, a_t \sim \pi_{\mathrm{behav}}} \left[ \sum_{t=1}^{H} \frac{\pi_{\mathrm{prox}}}{\pi_{\mathrm{behav}}} \min\left( u_t^{\mathrm{prox}}(\theta) \hat{A}_t, \mathrm{clip}\left( u_t^{\mathrm{prox}}(\theta), 1-\epsilon, 1+\epsilon \right) \hat{A}_t \right) \right],
\tag{5}
$$

where $u_t^{\mathrm{prox}}(\theta) = \frac{\pi_\theta(a_t \mid s_t)}{\pi_{\mathrm{prox}}(a_t \mid s_t)}$ is the importance ratio with respect to the proximal policy. We omit the state-action terms for conciseness.

The main difference between the asynchronous PPO objective in Equation 5 and the standard one in Equation 2 lies in the proximal policy $\pi_{\mathrm{prox}}$ for regularizing the model update. In asynchronous PPO training, using the behavior policy as the proximal policy will pull the latest policy $\pi_\theta$ towards the old-version and low-quality policies, thus slowing down model improvements. By employing a recent policy as the proximal policy, model updates happen within the trust region around the high-quality proximal policy $\pi_{\mathrm{prox}}$, thus stabilizing training.

The decoupled PPO objective in Equation 5 provides a natural benefit: it relaxes the requirement that all data within one training batch should be generated with a single policy. This property is crucial for maintaining algorithmic correctness when combining interruptible generation with policy updates. We claim that the inconsistent policy versions across a trajectory maintain equivalence to a single behavior policy $\pi_{\mathrm{behav}}$. (See Section D for the proof.)

**Proposition 1.** *For any sequence* $(q, a_1, \ldots, a_H)$ *generated by policies* $(\pi_\theta, \ldots, \pi_{\theta+k})$ *where* $\pi_{\theta+i}$ *produces tokens* $(a_{t_i}, \ldots, a_{t_{i+1}})$, *where* $1 = t_0 < \cdots < t_{k+1} = H$, *there exists a behavior policy* $\pi_{\mathrm{behav}}$ *such that the interrupted generation is equivalent to sampling entirely from* $\pi_{\mathrm{behav}}$.

**Practical Remark** While Hilton et al. [10] maintains an exponential moving average of parameters for $\pi_{\mathrm{prox}}$, this approach is prohibitively expensive for LRMs. Consequently, we simply use the parameters before each model update step as $\pi_{\mathrm{prox}}$. Equation 5 is implemented by recomputing token probabilities upon the arrival of the global batch in each training step.

## 5 Implementation

We implement AREAL using Python and PyTorch [35] built upon the ReaLHF [27] framework. Our system combines SGLang [63] v0.4.6 for generation serving with Megatron-Core [46] v0.11.0 as the training backend, managed by SLURM [59] for resource scheduling. To maximize throughput for both generation and training phases, we implement several key system-level optimizations that address critical bottlenecks in the pipeline.

AREAL decouples GPU computation from CPU operations, including rule-based reward computation (such as string matching for math problems or unit test execution for code) and TCP-based data transfer. By executing these operations in separate threads and pipelining the workflow, we overlap reward computation and data transfer with subsequent generation requests. We use asyncio coroutines to concurrently run multiple requests in the rollout worker to avoid mutual blocking waits.

To handle training with variable-length sequences, we employ a padding-free sequence packing strategy coupled with a dynamic allocation algorithm. The algorithm balances token distribution across micro-batches under fixed memory constraints (see Algorithm 1). This approach maximizes GPU memory utilization while minimizing the number of required forward-backward passes.

# 6  Experiments

Our evaluation comprises three components: (1) comprehensive comparisons against state-of-the-art open-source frameworks across model sizes, (2) strong-scaling analysis with varying compute resources, and (3) ablation studies validating our design choices.

## 6.1  Experiment Setup

We evaluate AREAL on challenging math and coding tasks. We employ the distilled Qwen2 model series [54, 55] from DeepSeek-R1 [6] as base models (i.e., R1-Distilled-Qwen), spanning from 1.5B to 32B parameters. For each task-model combination, we train for a fixed number of PPO updates and evaluate the final checkpoint. Our evaluation of mathematical tasks follows the Qwen evaluation protocol [56, 13], while coding models are assessed on LiveCodeBench (8/1/24-2/1/25) [14] using the official protocol. Unless otherwise specified, we set the maximum staleness $\eta = 4$ for coding and $\eta = 8$ for math, and adopt the training configurations used in Section 6.2, with additional hyperparameters detailed in Appendix B.

We conduct experiments on an H800 GPU cluster comprising 64 nodes, each equipped with 8 GPUs. The cluster features NVLink for intra-node connectivity and RoCE with 3.2Tbps bandwidth for inter-node communication. To ensure rapid convergence, we allocate a minimum of 16 nodes as a baseline pod configuration for complete experiments. We scale the number of nodes proportionally with model size, ultimately utilizing 48 nodes for training our largest 32B parameter model. This scaling strategy enables us to run experiments of varying sizes in parallel while maintaining efficient resource utilization.

For AREAL, we maintain a fixed ratio between inference and training devices, allocating three-quarters of the devices for inference. This configuration was selected over an equal 50-50 partition based on our early experiments, where the 75-25 partition demonstrated higher training throughput. Although we adopt this ratio as a heuristic configuration, we emphasize that the optimal partition may vary across different settings and could potentially benefit from dynamic adjustment during training, as discussed in Section 7.

## 6.2  End-to-End Comparison

We establish two state-of-the-art baselines using synchronous RL systems: DeepScaleR [25] for mathematical reasoning with a 1.5B model, and DeepCoder [24] for code generation with a 14B model, both trained using verl [45]. For larger 7B and 32B models where comparable baselines are unavailable, we performed controlled experiments by training from scratch using a synchronous variant of AREAL. After training, we evaluate on the challenging AIME24 benchmark for math models and the LiveCodeBench [14] benchmark for coding models. Evaluation results on additional benchmarks are presented in Appendix C.

Our main results are shown in Table 1. Since the code for obtaining previous SOTA models can be out-of-date, we measure the throughput and estimate the training hours using the latest verl code for a fair comparison. AREAL consistently matches or exceeds baseline performance while achieving significant speedups without performance degradation. In particular, the end-to-end training time can be reduced by $2.77\times$ compared with synchronous systems.

## 6.3  Scalability

We compare the scalability of AREAL with verl [45], the state-of-the-art synchronous RL system, across different model sizes and context lengths. We select the minimum number of GPUs when verl does not encounter the OOM issue for 7B models and 32k context length, then we proportionally adjust the number of GPUs according to the model size. We measure the *effective throughput* for training, defined as the rate of consuming generated tokens during PPO updates, after proper warmup steps. Figure 4 presents the results for context lengths of 16k and 32k. Here, context length refers to the sum of prompt length and generated length, with the maximum prompt length capped at 1k.

Across all settings, AREAL demonstrates an approximate linear scaling trend with increased device count, while the synchronous system typically fails to scale effectively. AREAL's throughput surpasses the baseline in most settings, and could achieve at most $2.5\times$ speedup. We note that for

Table 1: End-to-End Performance Comparison. We evaluate on the AIME24 benchmark for math and LiveCodeBench (8/1/24-2/1/25) for coding. We limit the maximum generation length to 32K tokens and sample 32 responses per question, reporting the average pass@1 accuracy. * represents the best known reproducible results obtained via RL, as cited from DeepScaler [25] and DeepCoder [24] respectively. AReaL achieves comparable performance with 2× fewer training hours.

| Model | AIME24 ↑ | # Nodes | PPO Steps | Training Hours ↓ |
|---|---|---|---|---|
| 1.5B basemodel | 29.3 | - | - | - |
| w/ VeRL | **43.1*** | 16 | 250 | 33.6 |
| w/ Sync.AReaL | 42.0 | 16 | 250 | 41.0 |
| w/ AReaL (ours) | 42.2 | 16 | 250 | **14.8** |
| 7B basemodel | 54.3 | - | - | - |
| w/ VeRL | - | 24 | 250 | 52.1 |
| w/ Sync.AReaL | 63.0 | 24 | 250 | 57.7 |
| w/ AReaL (ours) | **63.1** | 24 | 250 | **25.4** |

| Model | LiveCodeBench ↑ | # Nodes | PPO Steps | Training Hours ↓ |
|---|---|---|---|---|
| 14B basemodel | 53.4 | - | - | - |
| w/ VeRL | 57.9* | 32 | 80 | 44.4 |
| w/ Sync.AReaL | 56.7 | 32 | 80 | 48.8 |
| w/ AReaL (ours) | **58.1** | 32 | 80 | **21.9** |
| 32B basemodel | 57.4 | - | - | - |
| w/ VeRL | - | 48 | 60 | 46.4 |
| w/ Sync.AReaL | **61.2** | 48 | 60 | 51.1 |
| w/ AReaL (ours) | 61.0 | 48 | 60 | **31.1** |

smaller context lengths, the advantage of AREAL can be smaller because the generation throughput cannot match the pace of training throughput. Although many sequences are generated, they are not effectively consumed by the training process. Additionally, AREAL is more robust with longer generation lengths due to asynchronous and interruptible generation. The generation of long responses can be fully hidden in the critical path, so extending generation length does not drastically affect the effective training throughput of AREAL.

## 6.4 Algorithm Ablations

We conduct ablation studies to validate our algorithmic innovations in Section 4 by training a 1.5B LRM on math tasks. We follow the basic experiment setting of DeepScaleR and then gradually

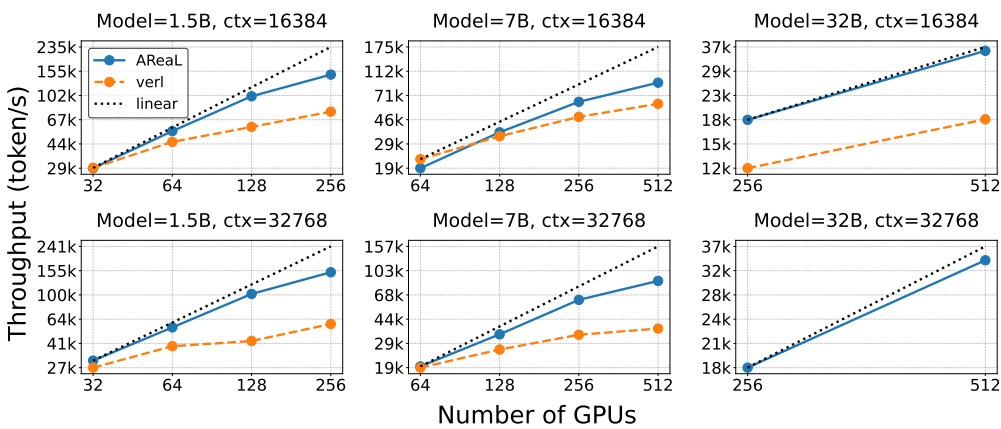

Figure 4: The strong scaling trend. Dotted lines indicate ideal linear scaling. verl consistently encounters OOM with 32k context length and the 32B model so the data points are missing.

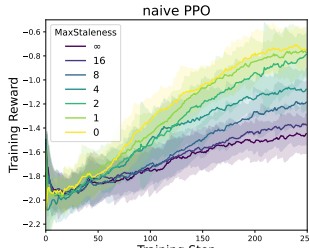
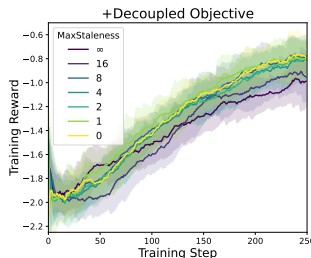
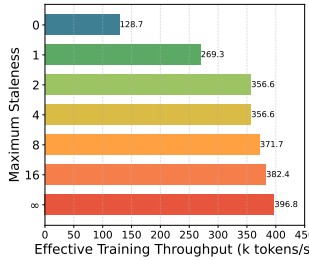

(a) Learning curves with naive PPO.

(b) Learning curves with eq. (5).

(c) Effective training throughput.

Figure 5: Ablation studies of the decoupled PPO objective and staleness control with a 1.5B model on math reasoning tasks. Both algorithmic choices are essential. With a moderate staleness value and the decoupled objective, training progress can be accelerated by over $2\times$ while maintaining final evaluation performance.

Table 2: Evaluation scores when varying data staleness, comparing performance with and without the decoupled objective. Numbers within $\pm 1$ of the oracle score are underlined.

| Max.Stale. | AIME24 | | AIME25 | | AMC23 | | MATH 500 | |
|---|---|---|---|---|---|---|---|---|
| | W/o | With | W/o | With | W/o | With | W/o | With |
| 0 (Oracle) | 42.0 | | 32.9 | | 84.4 | | 89.2 | |
| 1 | 41.8 | 42.1 | 30.7 | 31.9 | 83.3 | 85.2 | 89.9 | 89.8 |
| 2 | 40.0 | 41.8 | 32.1 | 32.5 | 82.3 | 84.3 | 89.6 | 89.6 |
| 4 | 23.3 | 42.2 | 23.1 | 32.0 | 58.5 | 85.1 | 66.9 | 89.5 |
| 8 | 35.7 | 41.0 | 27.8 | 31.1 | 81.2 | 82.9 | 87.8 | 89.2 |
| 16 | 35.8 | 38.7 | 26.2 | 32.5 | 78.4 | 83.2 | 87.4 | 89.1 |
| $\infty$ | 34.0 | 36.9 | 26.9 | 29.9 | 79.4 | 81.0 | 87.1 | 88.1 |

increase the $\eta$ value for ablation purposes. Specifically, we vary the maximum allowed staleness $\eta$ and compare configurations with and without the decoupled PPO objective. Figures 5a and 5b show the learning curves after 250 training steps. Table 2 presents the corresponding final evaluation performances across multiple mathematical reasoning benchmarks. We follow the common practice of PPO and perform multiple mini-batch updates within each training step. We emphasize that $\eta$ constrains the training batch staleness regarding training steps.

Figure 5a demonstrates that naive PPO fails to match the performance of the synchronous RL oracle (i.e., the performance when $\eta = 0$). Even slight staleness can significantly degrade final performance due to the improper clipping center and policy changes during interruptible generation. Furthermore, increasing data staleness consistently degrades learning performance, aligning with observations from prior work in other domains [2, 30]. However, as shown by comparing Figure 5b and Figure 5a, the decoupled PPO objective substantially improves training stability when handling stale data, consistent with findings from [10] in game domains. In addition, we observe that even with the decoupled objective, unbounded staleness (maximum staleness $\to \infty$) still results in inferior performance compared to the zero-staleness oracle. When properly constrained, moderate staleness (e.g., $\eta \leq 8$) has minimal impact on final performance while significantly accelerating training through the asynchronous pipeline, as demonstrated in Figure 5c and Table 2. These results validate our approach of combining controlled staleness with the decoupled PPO objective for efficient asynchronous RL training.

## 6.5 System Ablations

**Dynamic Microbatch Allocation** We investigate the effectiveness of dynamic batching by comparing PPO training throughput against a standard micro-batching strategy. The standard micro-batching strategy can result in multiple long sequences being assigned to the same micro-batch, thus usually requiring a sufficiently large number of micro-batches to prevent out-of-memory errors. In our

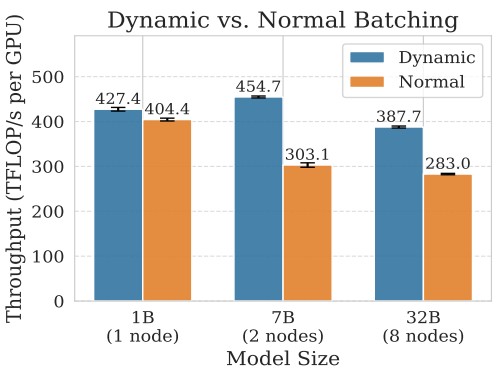
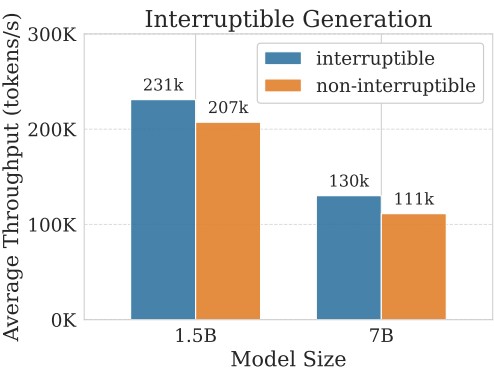

(a) Ablation study of dynamic micro-batch allocation.

(b) Ablation study of interruptible generation.

Figure 6: Ablation studies on system optimizations.

experimental setup, we configured 32 micro-batches for the standard setting and established a token budget of 32,768 per micro-batch for the dynamic batching approach. As demonstrated in Figure 6a, dynamic batching yields an average of 30% throughput improvements across various model sizes.

**Interruptible Generation** We ablate interruptible generation and present the resulting generation throughput in Figure 6b. Without interruptible generation, the controller must wait for the longest response. In particular, interruptible generation leads to a 12% and 17% throughput increase for 1.5B and 7B models respectively on 4 nodes, which validates our architectural design choice.

## 7 Conclusion

This paper introduces AREAL, a fully asynchronous system designed for efficient large-scale re-inforcement learning (RL) training. The AREAL architecture provides both the flexibility and expressiveness required for implementing asynchronous algorithms. Building upon this foundation, we contribute several algorithmic innovations, including staleness-aware training and a decoupled PPO objective, which enable efficient and stable PPO training in asynchronous environments. Our experimental results demonstrate AREAL's superior hardware efficiency, sample efficiency, and scalability compared to existing synchronous RL systems. This work provides a starting point for reliably scaling RL training. We hope that it can enable future advances in large-scale AI systems that push the boundaries of machine intelligence further.

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

# A Reproducibility

The code of AREAL is available at `https://github.com/inclusionAI/AReaL/`. Datasets and base models in our experiments are all taken from the open-source community (see Appendix B). We used a fixed random seed of 1 across all experiments.

# B Implementation Details

## B.1 PPO Details

We disable the critic model and the reference model in PPO. The advantage estimation parameter $\lambda$ in GAE and the RL discount factor $\gamma$ are fixed at 1. The reward is 5 at the final token if the answer is correct and -5 otherwise. We additionally adopt advantage normalization across the global batch to stabilize the training. Other learning related hyperparameters and configurations can be found in Table 3.

Table 3: Training configurations and hyperparameters.

| **Training Configuration** | |
| --- | --- |
| Batch size (number of prompts) | 512 |
| Random seed | 1 |
| **PPO Parameters** | |
| PPO Minibatches | 4 |
| Clipping $\epsilon$ | 0.2 |
| Advantage normalization | True |
| Discount factor $\gamma$ | 1.0 |
| GAE $\lambda$ | 1.0 |
| **Optimizer Parameters** | |
| Optimizer | Adam |
| Learning rate | $2.0 \times 10^{-5}$ |
| Weight decay | 0.05 |
| $\beta_1$ | 0.9 |
| $\beta_2$ | 0.95 |
| Adam $\epsilon$ | $1 \times 10^{-5}$ |
| Gradient norm clipping | 1.0 |
| Learning rate scheduler | constant |
| Warmup steps proportion | 0.001 |
| **Precision Parameters** | |
| Parameter dtype | fp16 |
| KV cache dtype | fp16 |
| Gradient dtype | fp32 |
| Optimizer state dtype | fp32 |
| **Generation Parameters** | |
| Answers per prompt | 16 |
| Temperature | 1.0 |
| Top-p | 1.0 |
| Top-k | -1 |
| Max prompt length | 1024 |
| Min generation length | 0 |
| Max generation length | 27648 |

## B.2 Dataset Details

For the math task, we used the open-source data from DeepScaleR [25], For code training, we used the dataset released by DeepCoder [24]. All compared methods use the same dataset.

## B.3 Dynamic Batching

---
**Algorithm 1** Dynamic Batching
---
**Require:** Sequence lengths $S = \{s_1, s_2, \ldots, s_n\}$, maximum micro-batch capacity $C$, minimum number of micro-batches $k_{min}$
**Ensure:** Balanced partition of sequences into micro-batches with total length $\leq C$
 1: Sort $S$ in descending order
 2: batches $\leftarrow \emptyset$
 3: **for all** $s \in S$ **do**
 4:     **if** $|\text{batches}| < k_{min}$ **or** no existing batch can fit $s$ **then**
 5:         Create new micro-batch containing sequence $i$
 6:         batches.append($\{s\}$)
 7:     **else**
 8:         Find all batches that can accommodate $s$
 9:         Select the micro-batch with fewest sequences
10:     **end if**
11: **end for**
12: **return** batches
---

The dynamic batching algorithm is shown in Algorithm 1.

## B.4 Baselines

In our experiments, we use the lastest version (main branch of verl repository, May 7, 2025) of verl [45] to evaluate the training throughput in Figure 4 and the training hours in Table 1. For most of the results, we use SGLang [63] v0.4.6 as generation backend and pytorch FSDP [62] as training backend. In a few cases where SGLang raises errors (experiments with 32B models or 64 nodes), we use vLLM [18] v0.8.4 as a substitution.

# C Additional Results

## C.1 Additional Evaluation Results

We evaluate the models trained with AReaL on more math and coding benchmarks, and list the results in Table 4 and Table 5, respectively.

Table 4: Results on math benchmarks.

| Model | AIME24 | AIME25 | AMC23 | MATH 500 |
|---|---|---|---|---|
| 1.5B basemodel | 29.3 | 24.4 | 71.0 | 84.3 |
| w/ Sync. AReaL | 42.0 | 32.9 | 84.4 | 89.2 |
| w/ AReaL | 42.2 | 32.0 | 85.1 | 89.5 |
| 7B basemodel | 54.3 | 41.7 | 89.5 | 92.8 |
| w/ Sync. AReaL | 63.0 | 50.0 | 93.2 | 94.2 |
| w/ AReaL | 63.1 | 47.3 | 93.6 | 94.3 |

## C.2 Generalization Across Model Architectures

We conducted additional experiments using the DeepSeek-Distilled-Llama-8B model, which is a long-CoT model based on Llama 3.1 8B. We matched the experimental configuration with the Qwen 7B math model from Table 1, and the results are presented in Table 6.

Table 5: Results on coding benchmarks.

| Model | LiveCodeBench v5 | Codeforces | CodeContests |
|---|---|---|---|
| 14B base model | 53.4 | 1801/95.8% | 32.0 |
| Sync. AReaL 14B | 56.7 | 1845/96.4% | 37.0 |
| AReaL 14B (ours) | 58.1 | 1840/96.3% | 35.9 |
| 32B base model | 57.4 | 1839/96.3% | 34.3 |
| Sync. AReaL 32B | 61.2 | 1911/96.9% | 36.3 |
| AReaL 32B (ours) | 61.0 | 1889/96.7% | 36.5 |

Table 6: Generalization results on DeepSeek-Distilled-Llama-8B across math benchmarks.

| Model | AIME24 | AMC23 | MATH500 | AIME25 |
|---|---|---|---|---|
| DeepSeek-Distilled-Llama-8B | 50.4 | 84.2 | 89.1 | 23.3 |
| AREAL Fine-Tuned $\eta =4$ | 58.4 | 92.3 | 92.2 | 42.6 |
| AREAL Fine-Tuned $\eta =8$ | 57.2 | 91.5 | 91.9 | 41.6 |

The results demonstrate that AReaL generalizes effectively across different model families.

### C.3 Staleness-Throughput Trade-off with Small-Scale Academic Setups

We conducted a series of experiments using the DeepSeek-Distilled-Qwen-1.5B model with 8k context length and batch size $64 \times 16$ on 8 GPUs, testing various staleness values. The following table shows the experimental results. We observe that our preliminary conclusions from the large-scale setting (Table 2) generally align with findings using fewer GPUs.

Table 7: Staleness-throughput trade-off on small-scale academic setup.

| Model | AIME24 | AIME25 | AMC23 | MATH500 | Throughput |
|---|---|---|---|---|---|
| DeepSeek-Distilled-Qwen-1.5B | 29.3 | 24.4 | 71.0 | 84.3 | - |
| AREAL Fine-Tuned $\eta =0$ | 31.7 | 26.1 | 78.9 | 86.7 | 27.1k |
| AREAL Fine-Tuned $\eta =1$ | 32.6 | 26.4 | 76.6 | 86.4 | 47.8k |
| AREAL Fine-Tuned $\eta =2$ | 32.4 | 26.7 | 76.0 | 86.6 | 47.8k |
| AREAL Fine-Tuned $\eta =4$ | 34.1 | 28.1 | 75.5 | 86.9 | 49.0k |
| AREAL Fine-Tuned $\eta =8$ | 29.9 | 23.2 | 76.1 | 86.1 | 51.5k |
| AREAL Fine-Tuned $\eta =16$ | 32.8 | 25.9 | 78.1 | 86.3 | 52.0k |

### C.4 Staleness-Throughput Trade-off with Different RL Algorithms

We conducted additional experiments using the RLOO advantage bootstrapping method. We trained 1.5B models in a local 8-GPU setting with 8k context length and various staleness values. The evaluation results in Table 8 demonstrate that RLOO exhibits slightly better tolerance to asynchronous training compared to vanilla PPO.

Table 8: Staleness-throughput trade-off using RLOO algorithm.

| Model | AIME24 | AIME25 | AMC23 | MATH500 | Throughput |
|---|---|---|---|---|---|
| DeepSeek-Distilled-Qwen-1.5B | 29.3 | 24.4 | 71.0 | 84.3 | - |
| RLOO $\eta =0$ | 32.4 | 29.2 | 79.2 | 87.3 | 27.1k |
| RLOO $\eta =1$ | 32.9 | 26.0 | 76.4 | 87.7 | 47.8k |
| RLOO $\eta =2$ | 34.1 | 28.0 | 81.1 | 86.9 | 47.8k |
| RLOO $\eta =4$ | 32.9 | 27.9 | 76.0 | 87.0 | 49.0k |
| RLOO $\eta =8$ | 31.5 | 28.1 | 78.0 | 87.4 | 51.5k |
| RLOO $\eta =16$ | 32.7 | 27.7 | 78.4 | 87.4 | 52.0k |

These results highlight an important direction for future research. AREAL modifies the PPO/GRPO algorithm because the importance sampling term naturally supports asynchronous off-policy training. Beyond PPO-based workflows, it would be valuable to investigate the asynchronous tolerance of REINFORCE-like and other off-policy algorithms.

## D  Proof of Proposition 1

**Proposition 1.** *For any sequence $(q, a_1, \ldots, a_H)$ generated by policies $(\pi_\theta, \ldots, \pi_{\theta+k})$ where $\pi_{\theta+i}$ produces tokens $(a_{t_i}, \ldots, a_{t_{i+1}})$, where $1 = t_0 < \cdots < t_{k+1} = H$, there exists a behavior policy $\pi_{\text{behav}}$ such that the interrupted generation is equivalent to sampling entirely from $\pi_{\text{behav}}$.*

*Proof.* For question $q$, let $\mathcal{S}_t(q)$ denote states encountered at step $t$ by the sequence of policies. Since $\mathcal{S}_{t_i}(q) \cap \mathcal{S}_{t_j}(q) = \emptyset$ for $i \neq j$, we can construct:

$$
\pi_{\text{behav}}(\cdot|s) = \begin{cases} \pi_{\theta+j}(\cdot|s) & \text{if } t_j \leq t \leq t_{j+1} \text{ and } s \in \mathcal{S}_t(q) \\ \text{arbitrary} & \text{otherwise} \end{cases}
$$

$\square$

## E  Limitations and Future Work

Our work presents several limitations that suggest directions for future research. First, the ratio between inference and training devices could be further optimized for specific training setups. Additionally, this ratio might benefit from dynamic adjustment during training, particularly as context lengths typically increase when fine-tuning pre-trained base models. While we focused our evaluation on single-step mathematical and coding tasks, the AREAL architecture is not inherently limited to these domains. We leave the exploration of multi-turn interactions and agentic scenarios to future work.

