# OpenReview forum: "AREAL: A Large-Scale Asynchronous Reinforcement Learning System for Language Reasoning"
_NeurIPS.cc/2025/Conference — NeurIPS 2025 poster_

### Official Review · Reviewer_ETFe · 2025-06-27

**Clarity:** 3
**Significance:** 3
**Originality:** 3
**Rating:** 5
**Confidence:** 4

**Summary:**

The authors present AReaL, a fully asynchronous system design enabling efficient RL for reasoning models. This system uses a “streaming”-like approach to LRM sampling that accounts for the higher & more heterogeneous inference workloads that come with reasoning. The authors stress test the asynchronous tolerance of standard PPO under their system and design a modified, more tolerant version of the learning objective accordingly. Experiments are conducted on the competitive math and programming settings.

**Questions:**

**Suggestions**

1. Extending the experiments in figures 5(a) and (b) to a few other extremely popular RL algorithm variants (e.g. GRPO & RLOO), as well as at least one model series other than Qwen would improve the experimental soundness of the work.

2. Nit: Many of the figures in the paper use different fonts, font size, and color schemes. While this is a minor detail, it is distracting for a reader.


**Questions**

> However, all these systems still follow a batched generation setting, where all the samples 50 within a training batch are from the same model version.

How exactly does AReaL completely avoid this? From what I understood, there is still micro-batching at the device level during generation -- could you please expound upon this point?

**Ethical Concerns:**

["NO or VERY MINOR ethics concerns only"]

**Final Justification:**

In their rebuttal, the authors:

(1) Expanded asynchronous tolerance evaluation with their framework from just PPO to RLOO, a popular alternative to training reasoning models.
(2) Added an experiment on asynchronous tolerance with a Llama model, expanding from Qwen-only evaluations.

I believe these additional experiments (especially (1)) strengthen the soundnesses of the work and its impact for the community.

Overall, reviewers agree that the work has good quality, clarity, significance, and originality. Thus, I raise my score to "accept" -- this is a good paper.

**Limitations:**

Yes.

**Paper Formatting Concerns:**

The limitations and future work section should be moved from the Appendix into the main body of the paper.

**Quality:**

3

**Strengths And Weaknesses:**

### Strengths
- The paper is well-written. The figures are relatively clean and easy-to-follow.
- The motivation for the method is compelling and the results are strong, though somewhat limited in scope. A new, fast, and open-sourced framework for RL is a valuable contribution to the community.

---

### Weaknesses

**Limited scope of experimentation**

Experiments with AReaL are limited to (1) PPO and (2) Qwen models.

(1) While PPO is perhaps the most standard evaluation choice, variants of PPO like GRPO and alternative algorithms like RLOO are also popular choices for training reasoning models in the community. It would be valuable to see (a) what the differences in asynchronous tolerance and (b) the AReaL tradeoffs between speedup vs stability & performance gain look like across algorithms with different variants of advantage bootstrapping.

(2) Many works have reported that reasoning-style RL on Qwen models behaves very differently to models from other families [1, 2]. Given this, I would be interested to see at least the experiments in Figure 5 (a) and (b) extended to another model family. For example, it is non-obvious to me that the degree of stability to off-policy data may not also be a somewhat model-specific property.


[1] Shao, R., Li, S. S., Xin, R., Geng, S., Wang, Y., Oh, S., ... & Zettlemoyer, L. (2025). Spurious Rewards: Rethinking Training Signals in RLVR. arXiv preprint arXiv:2506.10947.

[2] Gandhi, K., Chakravarthy, A., Singh, A., Lile, N., & Goodman, N. D. (2025). Cognitive behaviors that enable self-improving reasoners, or, four habits of highly effective stars. arXiv preprint arXiv:2503.01307.

---

> ### Author Rebuttal · Authors · 2025-07-31
>
> We sincerely appreciate your thorough and insightful review\! We would like to address your concerns with the following responses:
>
> > It would be valuable to see (a) what the differences in asynchronous tolerance and (b) the AReaL tradeoffs between speedup vs stability & performance gain look like across algorithms with different variants of advantage bootstrapping.
>
> We conducted additional experiments using the RLOO advantage bootstrapping method. Due to the tight rebuttal timeline, we ran 1.5B models in a local 8 GPU setting with 8k context length and various staleness values. The following evaluation results demonstrate that RLOO exhibits slightly better asynchronous tolerance to vanilla PPO.
>
> | model | AIME24 | AIME25 | AMC23 | MATH 500 | Throughput |
> | :---- | :---- | :---- | :---- | :---- | :---- |
> | 1.5B base | 29.3  | 24.4  | 71.0 | 84.3 | - |
> | RLOO staleness=0 | 32.4 | 29.2 | 79.2 | 87.3 |27.1k |
> | RLOO staleness=1 | 32.9 | 26.0 | 76.4 | 87.7 | 47.8k |
> | RLOO staleness=2 | 34.1 | 28.0 | 81.1 | 86.9 | 47.8k
> | RLOO staleness=4 | 32.9 | 27.9 | 76.0 | 87.0 | 49.0k |
> | RLOO staleness=8 | 31.5 | 28.1 | 78.0 | 87.4 | 51.5k |
> | RLOO staleness=16 | 32.7 | 27.7 | 78.4 | 87.4 | 52.0k |
>
> The reviewer raises an important direction for future research. AReaL modifies the PPO/GRPO algorithm because the importance sampling term naturally supports asynchronous off-policy training. Beyond PPO-based workflows, it would be valuable to investigate asynchronous tolerance for REINFORCE-like and other off-policy algorithms. We consider this an important avenue for future work.
>
> > Many works have reported that reasoning-style RL on Qwen models behaves very differently to models from other families \[1, 2\]. Given this, I would be interested to see at least the experiments in Figure 5 (a) and (b) extended to another model family.
>
> Thank you for this valuable suggestion\! We should clarify that there exists a paradigm difference: references \[1,2\] mainly investigate reasoning patterns starting from ***base or instruction-tuned*** models, while AReaL focuses on RL fine-tuning of ***long CoT models*** (e.g., distilled from OpenAI-o1/DeepSeek-R1). After distillation, the reasoning pattern should be less affected by the base model. To validate this point, we conducted additional experiments with DeepSeek-R1-Distill-Llama-8B in the math setting. The following results demonstrate that AReaL's effectiveness transfers across different model families:
>
> | model | aime24 | amc23 | math500 | aime25 |
> | :---- | :---- | :---- | :---- | :---- |
> | base model | 50.4 | 84.2 | 89.1 | 23.3 |
> | AReaL staleness=4 | 58.4 | 92.3 | 92.2 | 42.6 |
> | AReaL staleness=8 | 57.2 | 91.5 | 91.9 | 41.6 |
>
> We primarily used the Qwen family for experiments because: (1) it offers a range of model sizes (from 1.5B to 32B), which is convenient for both ablation studies (Figure 5\) and scalability validation of AReaL, and (2) the Qwen family provides readily available distilled long CoT models for RL training (i.e., DeepSeek-R1-Distill-Qwen). While running ablation studies on other model families (e.g., Llama 8B) is more computationally expensive, we are prepared to conduct these experiments when additional computational resources become available and will include these results in the camera-ready version.
>
> > How exactly does AReaL completely avoid batching?
>
> AReaL avoids generation batching within the RL pipeline, but the inference engine will automatically handle request batching on GPU. To be more specific, instead of calling batched generation with `engine.generate`, AReaL asynchronously sends individual generation requests to a remote HTTP server. The server (e.g., vLLM or SGLang) handles request batching according to its internal scheduler logic and executes batched computation on the GPU, then returns HTTP responses individually to the RL pipeline. Therefore, from the perspective of the RL pipeline, AReaL completely avoids batching during rollout while still benefiting from the server-side batching computations.

---

> > ### Comment · Reviewer_ETFe · 2025-08-02
> > **Response to Rebuttal**
> >
> > Thank you very much for the additional experiments with RLOO and DeepSeek-R1-Distill-Llama-8B despite the tight rebuttal deadline, as well as your clarification on the batching point!
> >
> > The rebuttal addresses my concerns -- I will increase my score to "accept".
> >
> > The experimental results with RLOO are especially interesting, and I would encourage the authors to add these to the main body of the paper.

---

### Official Review · Reviewer_EQw4 · 2025-06-29

**Clarity:** 3
**Significance:** 4
**Originality:** 3
**Rating:** 5
**Confidence:** 4

**Summary:**

Training large AI models for complex reasoning tasks is often slow because the system has to constantly switch between generating text and then training on that text, leaving powerful GPUs idle. This paper introduces AREAL, an approach that gets around this bottleneck by letting the generation and training processes run completely separately and at the same time. While this creates a challenge of the model learning from slightly "stale" data, AREAL incorporates a smart version of PPO to effectively handle these outdated samples. Ultimately, this asynchronous system provides a significant speed-up in training time, all while achieving comparable on reasoning benchmarks.

**Questions:**

This approach is clearly effective at a large scale. Could you discuss on how practical it would be to do research on these type of methods for smaller-scale experiments (4-8 GPUs)?

**Ethical Concerns:**

["NO or VERY MINOR ethics concerns only"]

**Final Justification:**

The authors have clarified my points. After reading other reviewer, i think the paper is ready for acceptance.

**Limitations:**

yes

**Paper Formatting Concerns:**

.

**Quality:**

3

**Strengths And Weaknesses:**

### Strengths
- **Targets a real bottleneck** – proposes a fully asynchronous actor-learner pipeline that keeps GPUs busy by overlapping generation, reward computation and training.
- **Extensive empirical study** – evaluates 1.5 B, 7 B, 14 B and 32 B-parameter models on four math datasets (AIME 24/25, AMC 23, MATH 500), one code dataset (LiveCodeBench), scales to 512 GPUs, and includes ablations on staleness, batching and interruptible generation.
- **Clear efficiency gains** – achieves 2.6-2.8 × higher training throughput than synchronous baselines while approximately maintaining performance.

### Weaknesses
- **Narrow coding coverage** – Table 1 reports results only on LiveCodeBench; broader code tasks remain untested.
- **Heavy hardware demand** – experiments rely on clusters of many GPUs, with no guidance for small-scale academic setups (< 16 GPUs) where the staleness/throughput trade-off may differ.

---

> ### Author Rebuttal · Authors · 2025-07-31
>
> We appreciate the reviewer's detailed and constructive feedback\! We would like to address your concerns one by one in the following responses:
>
> > Table 1 reports results only on LiveCodeBench.
>
> During the rebuttal period, we conduct additional evaluations in the coding domain and present the results below. We will include these results in the appendix of the camera-ready version.
>
> | Model | LiveCodeBench v5 | Codeforces | CodeContests |
> | :---- | :---- | :---- | :---- |
> | 14B base model | 53.4 | 1801/95.8% | 32.0 |
> | Sync. AReaL 14B | 56.7 | 1845/96.4% | 37.0 |
> | AReaL 14B (ours) | 58.1 | 1840/96.3% | 35.9 |
> | 32B base model | 57.4 | 1839/96.3% | 34.3 |
> | Sync. AReaL 32B | 61.2 | 1911/96.9% | 36.3 |
> | AReaL 32B (ours) | 61.0 | 1889/96.7% | 36.5 |
>
> > Experiments rely on clusters of many GPUs, with no guidance for small-scale academic setups (\< 16 GPUs) where the staleness/throughput trade-off may differ.
>
> Thank you for this important observation\! We conducted a series of experiments using the Qwen 1.5B distilled model with 8k context length and batch size 64×16 on 8 GPUs, testing various staleness values. The following table shows the experimental results. We observe that our preliminary conclusions from the large-scale setting (Table 2\) generally align with findings using fewer GPUs.
>
> | model | AIME24 | AIME25 | AMC23 | MATH 500 | Throughput |
> | :---- | :---- | :---- | :---- | :---- | :---- |
> | 1.5B base | 29.3  | 24.4  | 71.0 | 84.3 | - |
> | staleness=0 | 31.7 | 26.1 | 78.9 | 86.7 | 27.1k |
> | staleness=1 | 32.6 | 26.4 | 76.6 | 86.4 | 47.8k |
> | staleness=2 | 32.4 | 26.7 | 76.0 | 86.6 | 47.8k |
> | staleness=4 | 34.1 | 28.1 | 75.5 | 86.9 | 49.0k |
> | staleness=8 | 29.9 | 23.2 | 76.1 | 86.1 | 51.5k |
> | staleness=16 | 32.8 | 25.9 | 78.1 | 86.3 | 52.0k |
>
> We note that fine-tuning strong reasoning models with RL is typically resource-intensive \[1\], so training with 8 GPUs has to use smaller context lengths, and the final performance may not match that obtained from large-scale distributed training. We consider improving the efficiency of asynchronous RL training for smaller setups an important future direction.
>
> \[1\] https://github.com/giterinhub/DeepScaleR-1.5B-Preview
>
> > Could you discuss how practical it would be to conduct research on these types of methods for smaller-scale experiments (4-8 GPUs)?
>
> AReaL provides benefits for long CoT and agentic settings regardless of experimental scale. When trajectory length variance is high, synchronous systems can easily encounter resource underutilization issues while waiting for the longest rollout to complete. AReaL eliminates this idle time through continuous batching and rollout interruption, thereby improving overall throughput (Figure 3).
>
> Researchers may have concerns about the usability and customizability of asynchronous systems. However, the algorithmic implementation of AReaL does not differ significantly from synchronous counterparts. Users simply need to switch between synchronous and asynchronous rollout methods. The algorithm orchestration code can remain identical for both synchronous and asynchronous RL:
>
> ```
> dataloader = StatefulDataloader(dataset)
> for step in range(max_steps):
>     if async_training:
>         batch = rollout.prepare_batch(dataloader, workflow=workflow)
>     else:
>         data = next(dataloader)
>         batch = rollout.rollout_batch(data, workflow=workflow)
>     # Prepare training inputs
>     adv_batch = actor.compute_advantages_and_returns(batch)
>     batch['advantages'] = adv_batch['advantages']
>     # Execute PPO update
>     stats = actor.ppo_update(batch)
> ```
>
> This design ensures that researchers can easily adopt AReaL in their existing workflows with minimal code modifications.

---

> > ### Comment · Reviewer_EQw4 · 2025-08-07
> >
> > Thank you for your response and for running the extra tests. After reviewing everything, I’ve decided to raise my score to 5.

---

### Official Review · Reviewer_9Jfg · 2025-07-02

**Clarity:** 3
**Significance:** 3
**Originality:** 3
**Rating:** 5
**Confidence:** 3

**Summary:**

This paper introduces AREAL, a fully asynchronous reinforcement learning (RL) system designed to improve the training efficiency of large reasoning models (LRMs), such as language models used for math and code reasoning tasks. Unlike existing synchronous RL frameworks that alternate between generation and training phases and suffer from GPU underutilization, AREAL decouples these two phases entirely. Generation and training run in parallel across dedicated GPU clusters, enabled by interruptible rollouts and a staleness-aware design.
To address the challenges introduced by data staleness and inconsistent model versions in an asynchronous setting, the authors propose algorithmic innovations including a decoupled PPO objective and bounded staleness control, ensuring training stability even when using partially outdated rollouts. The system further incorporates engineering optimizations like dynamic micro-batch allocation and parallelized reward computation.
Extensive experiments on math and coding benchmarks with Qwen models up to 32B parameters demonstrate that AREAL achieves up to 2.8× speedup over state-of-the-art synchronous RL systems while matching or surpassing performance in terms of task accuracy. The results highlight the promise of asynchronous RL systems for scalable and efficient LLM training.

**Questions:**

- A minor point at line 252. Section 8 -> Appendix E.

**Ethical Concerns:**

["NO or VERY MINOR ethics concerns only"]

**Final Justification:**

My concerns are largely resolved during the discussion stage.
This is a technically solid paper and could be impactful to open-source communities. The authors are also committed to develop more features to make this work adaptable to prevalent frameworks such as vLLM.

**Limitations:**

Yes

**Quality:**

3

**Strengths And Weaknesses:**

**Strengths:**
- RL is an important and attractive field in the community. The paper could be very appealing to the community as it introduces a fully asynchronous RL system for LLM training, addressing key limitations in synchronous systems such as GPU underutilization and poor scalability.
- AREAL is a well-rounded design in both senses of algorithm and system. The system design is well-motivated and the novels algorithms are also invented to solve the data staleness problem in a pretty natural and straightforward way.
- The empirical results of AREAL is strong. It achieves 2x throughput improvement while maintaining or even improving the performance on math and coding datasets.

**Weaknesses:**
- Despite the strong system and algorithmic contributions, the paper does not release any code. Given the paper's claim to be a infra contribution by offering a new paradigm for RL training, the absence of code will limits reproducibility and may hinder broader adoption and feedback from the open-source community.
- All experiments are conducted using only the Qwen family of models. This raises concerns about the generalizability of AREAL across different model architectures or training stacks (e.g., LLaMA, Gemma, Mistral).
- AREAL currently only supports SGLang as the generation backend, which may restrict its usability in broader settings. Since many research and industry users rely on frameworks like vLLM for generation, the system’s current reliance on SGLang may act as a barrier to integration and adoption.
- Some system-level design choices, such as the fixed ratio between inference and training, are based on heuristics without dynamic tuning or adaptation. A more adaptive approach could further improve throughput and robustness in varied deployment scenarios.

---

> ### Author Rebuttal · Authors · 2025-07-31
>
> Thank you for the detailed and insightful review\! We would like to address your concerns with the following responses:
>
> > The paper does not release any code.
>
> We have provided an anonymous link in Appendix A. We apologize that we cannot repost the link in this rebuttal response, as it is strictly prohibited by the rebuttal guidelines.
>
> > Concerns about the generalizability of AReaL across different model architectures.
>
> During the rebuttal period, we conducted additional experiments using the DeepSeek-Distilled-Llama-8B model, which is a long-CoT model based on Llama 3.1 8B. We matched the experimental configuration with the Qwen 7B math model from Table 1, and the results are presented in the following table:
>
> | model | aime24 | amc23 | math500 | aime25 |
> | :---- | :---- | :---- | :---- | :---- |
> | base model | 50.4 | 84.2 | 89.1 | 23.3 |
> | AReaL staleness=4 | 58.4 | 92.3 | 92.2 | 42.6 |
> | AReaL staleness=8 | 57.2 | 91.5 | 91.9 | 41.6 |
>
> The results demonstrate that AReaL generalizes effectively across different model families.
>
> > The system's current reliance on SGLang may act as a barrier to integration and adoption.
>
> We primarily incorporated and modified SGLang to implement the interruptible rollout workers, but AReaL is not restricted to specific inference and training backends. We commit to supporting additional backends in the future, including PyTorch FSDP for trainer and vLLM for rollout inference worker. This expansion is currently part of our ongoing internal development efforts and will be uploaded to an open-source repository in the future.
>
> > Some system-level design choices, such as the fixed ratio between inference and training, are based on heuristics without dynamic tuning or adaptation. A more adaptive approach could further improve throughput and robustness in varied deployment scenarios.
>
> The reviewer's observation is absolutely correct\! We currently use a heuristically tuned inference-to-training ratio designed to saturate training GPUs. This ratio could be further optimized (decreased) based on specific training configurations and would benefit significantly from dynamic adjustment during training, particularly when context lengths change during RL training.
> We will discuss these limitations in the appendix and consider these important ideas for future work. Thank you again for this insightful suggestion\!

---

> > ### Comment · Reviewer_9Jfg · 2025-08-03
> >
> > Thanks for your response and for conducting the additional experiments. I will maintain my score as it is.

---

### Official Review · Reviewer_u73n · 2025-07-03

**Clarity:** 3
**Significance:** 3
**Originality:** 3
**Rating:** 5
**Confidence:** 2

**Summary:**

This paper introduces AReaL, an asynchronous reinforcement learning system designed to efficiently train large language models (LLMs) on reasoning tasks. Unlike traditional RL systems that operate synchronously, forcing model training to wait for all generated outputs to complete, AReaL fully decouples generation from training. Rollout workers continuously produce new outputs without blocking, while training workers update the model as soon as data is ready, significantly improving GPU utilization. AReaL is a fully asynchronous RL system for training large language models on reasoning tasks. It decouples generation and training to improve GPU utilization and efficiency. With staleness-aware PPO and system optimizations, AReaL achieves up to 2.57× speedup over synchronous baselines while maintaining or improving performance.

**Questions:**

1.The strength of this paper is evident—AReaL significantly accelerates RL training. However, I noticed that many data points for VeRL are missing or marked with “-”. Since the comparison with VeRL is important (despite AReaL’s clear speed advantage), it would be helpful if the authors could include results across multiple random seeds. This would provide a clearer picture of how VeRL’s final performance compares to AReaL.

2.Additionally, from the reported results, it seems that VeRL occasionally outperforms AReaL. Does this imply that training on the most recent data may yield better model performance than using stale samples, at least within the current implementation?

**Ethical Concerns:**

["NO or VERY MINOR ethics concerns only"]

**Final Justification:**

I would like to keep my current positive rating of the paper, and encourage the authors to incorporate our discussions into the manuscript.

**Limitations:**

Yes

**Quality:**

4

**Strengths And Weaknesses:**

Strength:

1. The introduction of a decoupled PPO objective and staleness-aware training effectively tackles theoretical issues such as policy mismatch and data staleness—common bottlenecks in asynchronous RL. The paper provides formal analysis and empirical ablation to validate their approach.

2. AREAL is extensively evaluated on multiple benchmarks (AIME24, LiveCodeBench, MATH 500) using models up to 32B parameters. It consistently matches or exceeds prior methods while significantly reducing training time. The scalability results show near-linear throughput improvements up to 512 GPUs.


Weakness:

As a reviewer not familiar with system design, I didn’t identify any clear weaknesses in this paper. Please refer to my question for further clarification.

---

> ### Author Rebuttal · Authors · 2025-07-31
>
> We sincerely thank the reviewer for their supportive comments\! We hope the following responses address your concerns:
>
> > I noticed that many data points for VeRL are missing or marked with "-" in Table 1
>
> We acknowledge that despite our best efforts to tune VeRL before paper submission, our VeRL reproduction did not achieve comparable results to AReaL in some cases. Consequently, in the draft, we included results from third-party forks of VeRL, such as DeepScaleR\[1\] and DeepCoder\[2\], which typically incorporate customized modifications to improve training stability.
>
> Currently, we are re-running experiments with the latest version of VeRL to address these missing data points. However, given the computational expense of these experiments (e.g., the 7B math model requires over 10,000 GPU hours to complete), we have not been able to produce valid results during the rebuttal period. We will make every effort to include these amended results in the camera-ready version.
>
> \[1\] [https://huggingface.co/agentica-org/DeepScaleR-1.5B-Preview](https://huggingface.co/agentica-org/DeepScaleR-1.5B-Preview)
>
> \[2\] [https://www.together.ai/blog/deepcoder](https://www.together.ai/blog/deepcoder)
>
> > It would be helpful if the authors could include results across multiple random seeds. This would provide a clearer picture of how VeRL's final performance compares to AReaL.
>
> We completely agree that including multiple random seeds would strengthen the robustness of our results. In the current draft, AReaL uses a fixed random seed of 1 for all training experiments. The evaluation protocol samples 32 responses with different random seeds and reports the average accuracy, following common practices adopted in the community \[3\].
>
> We plan to re-run experiments with multiple random seeds to test AReaL's robustness once we have available computational resources. We commit to updating these results in the camera-ready version.
>
> \[3\] https://github.com/QwenLM/Qwen2.5-Math
>
> > VeRL occasionally outperforms AReaL. Does this imply that training on the most recent data may yield better model performance than using stale samples, at least within the current implementation?
>
> We agree with the reviewer about this observation. In general, training with more recent data does yield better performance. However, the algorithmic modifications in AReaL increase tolerance to staleness, enabling us to achieve comparable performance even with staler data.
>
> We note that the end-to-end comparison between VeRL and AReaL may not clearly showcase the effect of data staleness due to numerous implementation differences, such as the use of log-probabilities produced by the inference engine and different training/inference backends. Each of these factors can introduce numerical issues that influence final performance. Based on Table 1, we conclude that VeRL and AReaL demonstrate comparable algorithmic performance.
>
> Regarding the effect of data staleness, Table 2 presents a more controlled ablation study. The results show that staleness levels of 1, 2, and 4 all achieve similar final performance to the synchronous oracle (staleness=0) across multiple evaluation benchmarks. These results demonstrate AReaL's increased tolerance to data staleness.

---

### Note · Authors · 2025-08-12

We thank all reviewers for their constructive feedback and engaging discussions. This summary aims to provide a comprehensive overview of the review process.

## Core Strengths

All reviewers recognized AReaL's fundamental contribution as **a fully asynchronous RL system that addresses critical scalability bottlenecks in training large reasoning models**. The reviewers consistently highlighted:

- **Addressing real bottlenecks**: AReaL introduces a fully asynchronous RL system for LLM training, addressing key limitations in synchronous systems such as GPU underutilization and poor scalability.
- **Significant efficiency gains**: Up to 2.8× speedup over synchronous baselines while maintaining algorithmic performance
- **Well-rounded algorithm-system co-design**: Novel staleness-aware PPO modifications that effectively handle data staleness and policy mismatch in a natural and straightforward way
- **Comprehensive evaluation**: Extensive experiments across multiple model sizes (1.5B-32B parameters), benchmarks (AIME, AMC, MATH, LiveCodeBench, and Codeforces/CodeContests during rebuttal), and scales (up to 512 GPUs)

## Addressed Concerns

1. **Model family dependence**: Initially raised concerns about model family dependence were addressed through additional experiments with DeepSeek-R1-Distill-Llama-8B, demonstrating AReaL's effectiveness across different architectures.

2. **Algorithm generalizability**: We performed additional experiments with other algorithms (i.e., RLOO), showing similar or improved asynchronous tolerance compared to vanilla PPO. This opens up an important potential direction for future research.

3. **Accessibility for academic settings**: We clarified that AReaL requires minimal code modifications from existing synchronous workflows and provided small-scale experiments (8 GPUs) to address concerns about the usability in academic settings.

## Epilogue

We would like to express our sincere gratitude for the support and engagement of all reviewers, ACs, and SACs. We are committed to updating our draft with the new results and discussions following the reviewers' valuable suggestions.

We will also open-source our code and maintain it consistently to support additional functionalities (e.g., vLLM, FSDP) and algorithms (e.g., RLOO). We hope that AReaL will be more than just a research paper, but rather a solid toolset that can be applied by beginners, researchers, and industrial users in the field of reinforcement learning.

---

### Decision · Program_Chairs · 2025-09-17

**Decision:**

Accept (poster)

**Comment:**

All reviewers recommend clear acceptance, highlighting the paper’s solid technical contributions, strong empirical validation, and clear impact on the LLM community.

Several concerns raised by the reviewers were adequately addressed in the rebuttal, including:
- Comparison with VeRL (reviewer `u73n`)
- Model family dependence (reviewer `9Jfg`, `ETFe`)
- Coding coverage and hardward demand (reviewer `EQw4`)
- Unclear technical details (reviewer `ETFe`)

The authors are recommended to include these experiments/ablations in the revision and release the code to benefit the community.
Given the uniformly positive reviews and its contributions to both research and practice, I recommend acceptance.